# Three-dimensional composition of the photoreceptor cone layers in healthy eyes using adaptive-optics optical coherence tomography (AO-OCT)

**Adrian Reumueller**[1], **Lorenz Wassermann**[1], **Matthias Salas**[2], **Markus Schranz**[1], **Valentin Hacker**[1], **Georgios Mylonas**[1], **Stefan Sacu**[1], **Wolfgang Drexler**[2], **Michael Pircher**[2], **Ursula Schmidt-Erfurth**[1], **Andreas Pollreisz**[1]*

**1** Department of Ophthalmology and Optometry, Medical University of Vienna, Vienna, Austria, **2** Center for Medical Physics and Biomedical Engineering, Medical University Vienna, Vienna, Austria

* andreas.pollreisz@meduniwien.ac.at

## Abstract

### Purpose

To assess the signal composition of cone photoreceptors three-dimensionally in healthy retinas using adaptive optics optical coherence tomography (AO-OCT).

### Methods

**Study population.** Twenty healthy eyes of ten subjects (age 23 to 67).

**Procedures.** After routine ophthalmological assessments, eyes were examined using AO-OCT. Three-dimensional volumes were acquired at 2.5° and 6.5° foveal eccentricity in four main meridians (superior, nasal, inferior, temporal). Cone densities and signal compositions were investigated in four different planes: the cone inner segment outer segment junction (IS/OS), the cone outer segment combined with the IS/OS (ISOS+), the cone outer segment tips (COST) and full en-face plane (FEF) combining signals from all mentioned cone layers. Additionally, reliability of a simple semi-automated approach for assessment of cone density was tested.

**Main outcome measures.** Cone density of IS/OS, IS/OS+, COST and FEF. Qualitative depiction and composition of each cone layer. Inter-rater agreement of cone density measurements.

### Results

Mean overall cone density at all eccentricities was highest at the FEF plane (21.160/mm$^2$), followed by COST (20.450/mm$^2$), IS/OS+ (19.920/mm$^2$) and IS/OS (19.530/mm$^2$). The different meridians and eccentricities had a significant impact on cone density, with lower eccentricity resulting in higher cone densities (p≤.001), which were highest at the nasal, then temporal, then inferior and then superior meridian. Depiction of the cone mosaic differed between all 4 layers regarding signal size and packing density. Therefore, different cone layers showed evident but not complete signal overlap. Using the semi-automated technique for counting of cone signals achieved high inter-rater reliability (ICC > .99).

**Data Availability Statement:** All relevant data are within the manuscript and its Supporting Information files.

**Funding:** The authors received no specific funding for this work. The funders had no role in study design, data collection and analysis, decision to publish, or preparation of the manuscript.

**Competing interests:** The authors have declared that no competing interests exist.

## Conclusions

In healthy individuals qualitative and quantitative changes in cone signals are found not only in different eccentricities and meridians, but also within different photoreceptor layers. The variation between cone planes has to be considered when assessing the integrity of cone photoreceptors in healthy and diseased eyes using adaptive optics technology.

## Introduction

Visualization of the retina and especially the photoreceptor layers has been and still is a major goal of ophthalmic retinal imaging. With the introduction of gene therapies for inherited retinal diseases such as retinitis pigmentosa as well as the exploration of gene therapies for other more prevalent diseases like neovascular age-related macular degeneration (AMD), the importance of improved and more detailed visualization of the retina is growing further. Conventional optical coherence tomography (OCT) is the current mainstay of retinal imaging and its performance has been vastly improved since the introduction of spectral-domain and swept-source technology. However, despite the availability of ultrafast light sources, the transversal resolution of current commercial OCT systems is still limited as aberrations of the eye distort the wavefront of the imaging beam.

Adaptive optics (AO) is a technology that allows to correct for these aberrations and thus vastly increases the resolution of any imaging method in which it is implemented [1–4]. Currently, the most commonly used AO enhanced imaging technologies are AO flood illumination fundus camera (AO-FC) and AO scanning laser ophthalmoscopy (AO-SLO), which have both been applied on healthy eyes and in several retinal pathologies [5–11]. Although both technologies allow for good visualization of the photoreceptor cone mosaic, they are limited to a two dimensional en-face view. Thus, the en-face images from a specific retinal layer do also contain unwanted signals from other retinal layers. Additionally, the cone signals cannot be separated into the typical layers that we see in OCT B-Scans, the inner outer segment junction zone (IS/OS) and the cone outer segment end tips (COST), which can be relevant for assessment of photoreceptor impairment [12–16]. AO optical coherence tomography (AO-OCT) overcomes these limitations by achieving not only high lateral but also high axial resolution using partial coherence interferometry, thus acquiring full three-dimensional volumes and allowing for separation of the individual cone layers [2,3,11,17–21]. AO-OCT imaging has its limitations, as system set-ups are challenging, acquisition of volumes is more susceptible to motion artefacts and processing of volumes consumes significant resources. While different reflective properties of cone layers have been described with the introduction of AO-OCT [22–25], a systematic assessment of healthy individuals comparing signal densities of separate cone layers and evaluating the impact of the retinal meridians is still missing. We have developed an AO-OCT prototype that is optimized for clinical use [26], which has been recently applied in patients with AMD, retinal detachment and diabetic vasculopathy [27–29]. This study applies AO-OCT on healthy individuals to provide a closer look on the three-dimensional appearance of cones with AO-enhanced OCT, additionally answering the following questions: How is the quantitative relationship between photoreceptor signals in the different cone layers, and how do they compare to two-dimensional cone mosaic depiction of AO-FC/SLO in literature? And finally, how well do different graders agree in counting of AO-OCT photoreceptor signals when using a simple, semi-automated approach of counting?

## Methods

### Ethics approval and informed consent

This prospective case series using AO-OCT on healthy individuals was approved by the Medical University of Vienna institutional review board. The study was conducted between October 1st, 2018 and March 31st, 2019 and adhered to the tenets of the Declaration of Helsinki and the Guidelines for Good Clinical Practice. All participants received a full explanation of the study and gave written informed consent for participation.

### Subjects

20 eyes of 10 individuals were included. Study participants were recruited by public advertising. Possible candidates were any individuals age 18 to 70 without a past or present history of ocular diseases. To avoid factors decreasing image quality or altering cone morphology, individuals were excluded if they showed any of the following criteria: senile cataract above nuclear color (NC) 3.0 and/or nuclear opalescence (NO) 3.0 and/or cortical (C) 2.0 and/or posterior subcapsular (P) 0.1 using the Lens Opacities Classification System III, previous cataract surgery, insufficient pupil dilation below 5mm or significantly irregular pupil shape, insufficient ability to fixate, high myopia with spherical errors above 5.0 diopters or cylindrical error above 3.0 diopters and presence of any other media opacity or diagnosed ocular disease.

### Examinations

All patients received the following examinations at a single visit: testing of best corrected visual acuity (BCVA, Snellen chart, 6m), slit-lamp biomicroscopy including indirect fundus examination, AO-OCT [26], SD-OCT (Spectralis HRA+OCT, Heidelberg Engineering, Heidelberg, Germany) and measurement of axial length, anterior chamber depth and keratometry (IOL Master 500, Carl Zeiss Meditec, Jena, Germany). Imaging was performed after instillation of 1.0% tropicamide (Mydriaticum Agepha, Vienna, Austria) and 5.0% phenylephrine hydrochloride (Neosynephrin-POS 5%, Ursapharm, Saarbruecken, Germany) to achieve a minimum pupillary diameter of 5.0mm.

### AO-OCT assessment

AO-OCT volumes were acquired in both eyes of each subject at 8 regions: 2.5˚ of foveal eccentricity (ecc) superior, nasal, inferior and temporal and ecc 6.5˚ superior, nasal, inferior and temporal.

The AO-OCT system used was developed at the Center for Medical Physics and Biomedical Engineering of the Medical University Vienna, described [26,30] and clinically applied [27–30] previously. The AO-OCT system uses a superluminescent diode operating at 841nm for imaging in combination with a deformable mirror (Mirao52-e RC, Imagine Eyes, Orsay, France) and a Shack Hartmann wavefront sensor (Haso first, Imagine Optics, Orsay, France) for wavefront correction. An AO-FC fundus live display (Imagine Eyes, Orsay, France) is integrated for overview and an internal display serves as a fixation target for the patient to find the localization of interest. The correct positioning of the region of interest can be reassured comparing vessels as landmarks on the AO-OCT, AO-FC live display and conventional fundus images. The achieved optical resolution of the OCT is 4.5μm axially and 3.2μm laterally over a field of view of 2 x 2 degrees. One B-Scan consists of 365 A-Scans with an A-Scan rate of 200kHz and the acquisition time of a single volume consisting of 400 B-Scans takes approximately 800ms. The optical power for an assessment is 500μW at 841nm for the OCT imaging beam and additional 50μW at 750nm for the internal guide star (for wavefront sensing), which

is below the limits for safe exposure according to the European standard IEC 60825–1:2014. To ensure proper function of the AO correction, a numeric value (root mean square of the residual wavefront error) is shown in real-time to the operator, which has to be below a certain range (0.1μm) during volume acquisition. At least 3 volumes were acquired at each of the 8 regions. From these multiple volumes of each region, a single volume (which showed the least amount of micro saccades) was selected for further analysis. Volumes were processed using cross correlation to counteract axial eye movements as previously described [26,27] and then analysed in FIJI (Fiji-Is-Just-ImageJ 2.0.0-rc61/1.51n, https://fiji.sc/). If present, small tilts of the whole volume (usually between 0.1˚ and 2.0˚ clock- or counter-clockwise) were compensated by using the "rotate" function. A mean 3D filter (radius 1) was used for noise reduction. The volumes were analysed in orthogonal views, allowing for synchronous visualization of the x-, y- and z-plane. In every volume, "Z Project" function was used to create en-face images of the desired photoreceptor planes by projecting the corresponding en-face slices (each with a thickness of one pixel) to the final en-face image of each layer (such as IS/OS or COST). For "Z project", we used the "maximum intensity" type, as the aim was to find bright spots (cone signals) within the volume and not blurring out signals for smoother visual appearance which would be achieved by using the "mean average" or "median average" function. As each plane except the FEF consisted usually only of a few slices however, differences between the various "Z project" types were only subtle.

Assessment of cone density was performed over an area of 100x100pixels (approximately 160x160μm in an eye with 23mm length and 3.5mm AC depth). For counting of cones, we applied a simple semi-automated approach by first using the "Find Maxima" function in FIJI and then manually validating and correcting the results, as proposed by Zhang et al. [31] *Fig 1* shows an example how the "Find Maxima" function detects cone signals in a region of interest.

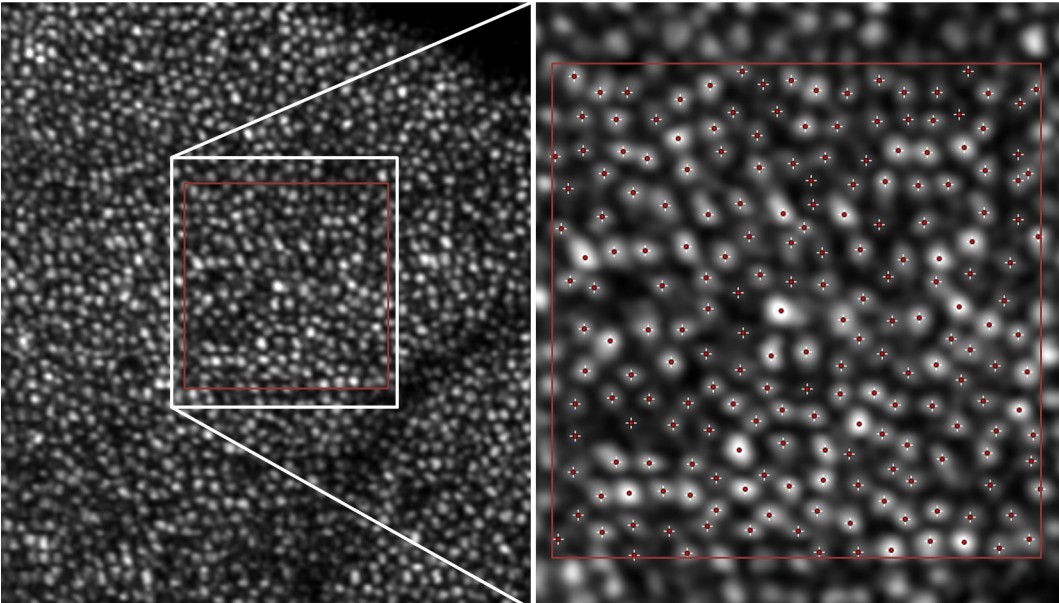

**Fig 1. Application of the "Find Maxima"-function in the open source software FIJI on a full en-face image depicting photoreceptors.** Shown is an example of photoreceptors at 6.5˚ foveal eccentricity inferior in a 23yo healthy woman. The region within the white rectangle is magnified on the right half of the image. The red rectangle depicts the region of interest. The "Find Maxima"-function determines and counts local maxima of intensity, offering a fast and easy way to find bright signals within an image, here indicated by red crosshairs. The examiner can add and remove selections freely. Using this function, even weaker reflective signals of cones can be detected and then validated, resulting in a total count of 205 cones for this image.

To assess the inter-rater reproducibility of this method for AO-OCT, one region was selected randomly from the IS/OS, the COST and the full en-face image of every eye (60 different images in total) and then analysed separately by three experienced graders (AR, MS, VH). We decided not to use images of the IS/OS+ layer for this comparison, as depiction of IS/OS + was very close to IS/OS.

## Statistics

We analyzed cone density values of 4 different layers (IS/OS, IS/OS+, COST and full en-face, described below in detail), in 4 meridians (superior, nasal, inferior, temporal) and two eccentricities (ecc 2.5˚ and ecc 6.5˚), resulting in a total of 640 measured areas. To asses if there was a difference in each density value between the right and the left eye, paired T-Test was used with a two sided p- value $< .05$ being considered significant. To asses if there was a difference between the density values between the layers, a general linear model ANOVA for repeated measurements was performed for each meridian and eccentricity with a $p < .05$ considered significant, reported either with assumed sphericity or Greenhouse-Geisser correction. If significant differences were found, pairwise comparisons were used for further assessment of relationships. The same method was applied to assess if there was a general difference in densities between the 4 meridians. For comprehensibility, only densities of full-en face images were used for this assessment, as these images showed the most complete appearance of the photoreceptor mosaic (see below). For assessment of inter-rater reliability, a two-way mixed effects model with absolute agreement was used.

## Results

Mean age of the 10 subjects (5 male, 5 female) was 38.5 years (min 23, max 67). Mean BCVA (decimal) was 1.16 (min 1.00, max 1.25), mean spherical equivalent -0.78 diopters (min -3.00, max +1.50), mean cylinder +0.64 diopters (min +0.00, max +2.00) and mean axial eye length 23.22mm (min 21.70, max 24.43). All eyes had clear media, were phakic and showed no relevant cataract.

Qualitative assessment of photoreceptor signals: In healthy individuals, the photoreceptors showed hyper-reflectivity at different planes in AO-OCT B-Scans, as shown in *Fig 2,* where the IS/OS, the cone outer segment (OS), the COST and the rod outer segment tips (ROST) are depicted.

The B-Scans of a volume allowed for segmentation of several en-face images: *Fig 3* shows a depiction of the complete retina (*CR plane*), which integrates signals from all retinal layers, starting with the ILM through the choroid. This image is comparable to what can be obtained with two-dimensional imaging methods such as AL-FC or AO-SLO. The cone signals, visible as hyper-reflective dots, have a lower contrast due to background signals, in particular from the ROST and the RPE. As a result, averaging of several acquisitions would be required to improve detectability. With AO-OCT however, the different layers forming the CR plane can be extracted as further shown.

The hyper-reflective signals from the IS/OS layer (*Fig 3, IS/OS plane*) form a mosaic of densely packed dots with oscillating reflectivity. Below the IS/OS plane, the cone outer segment plane (*Fig 3, OS plane*), can be visualized, which shows noticeably less signals than the IS/OS as only few cones have bold hyper-reflective dots whereas a few others show weak signals. Signals from the OS plane can fill signal gaps seen in the IS/OS plane, suggesting that some cones show their first bright signal not at level of the IS/OS layer but slightly below. Thus, we combined the signals from the IS/OS and OS layers to the composite *IS/OS+ plane* (*Fig 3*, second row). The cone mosaic of this image is formed mainly from IS/OS signals plus some additional

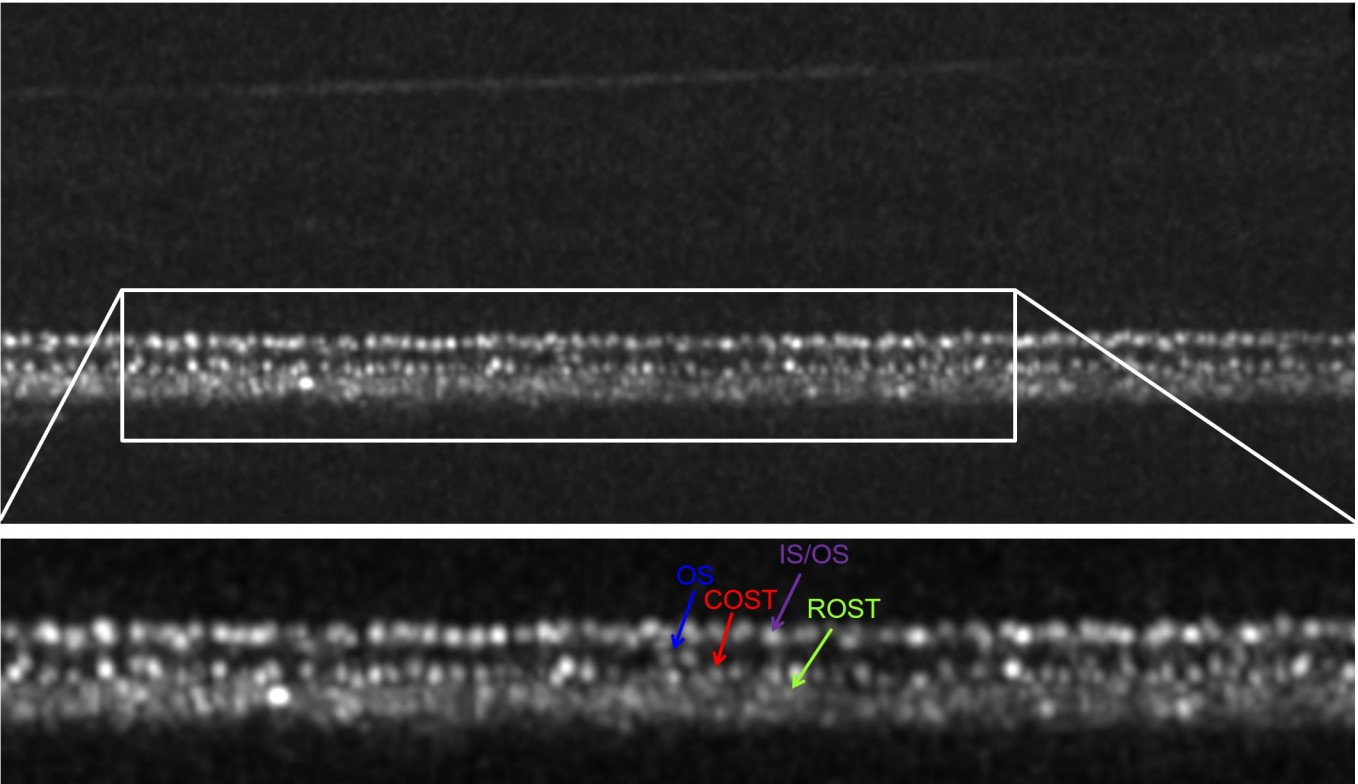

**Fig 2. AO-OCT B-Scan with labelled photoreceptor layers.** Non-averaged image, linear brightness scale, location 6.5˚ superior eccentricity in a 26yo healthy male. The upper image shows the whole B-Scan. Note, how the inner retinal layers are barely visible due to their lower reflectivity in comparison to the outer retinal layers. Visualization of the inner retina can be improved by focusing the AO-OCT on the nerve fibre layer instead of the RPE, switching from a linear to a logarithmic brightness scale and averaging multiple B-Scans to a single image. The white rectangle shows the region of interest of the outer retina, which can be seen in the lower section of the image: The cone inner segment-outer segment junction (**IS/OS**) consists of bold dots (purple arrow) with hyper reflectivity and with oscillating intensity, ranging from grey to bright white. Below the IS/OS, the cone outer segments (**OS**) show mostly very faint and blurry signals; a few hyper-reflective spots can be seen (blue arrow), most likely defects in the packing density of the cone outer segment discs (Pircher et al. 2011). Below the OS, the cone outer segment tips (**COST**) show a similar pattern as the IS/OS, but the spots (red arrow) appear slightly smaller and better separated. The rod outer segment tips (**ROST**), are located directly below the COST; depiction in a non-averaged image is difficult due to their small size, however, some signals can be depicted (green arrows). Note: Due to its weak reflectivity, the external limiting membrane, which is located above the IS/OS, can barely be seen in linear brightness scale and non-averaged images.

signals from the outer segments underneath. *S1 File* shows a comparison of the IS/OS and the IS/OS+ plane at ecc 2.5˚ nasal in a 27yo healthy male, depicting how signals from the OS plane partially "complete" the IS/OS plane.

Below the OS plane, a dense cone pattern of the *COST plane (Fig 3)* is found. Here, the hyper-reflective dots appear slightly smaller but therefore better separable compared to the IS/OS plane. Below the COST, the signal-network of the ROST and the signal patterns of the RPE are found, which were generally not further analyzed as they were not suitable for quantitative density evaluation because individual rod photoreceptors cannot be resolved with the system. However, *Fig 3* also shows an example of the *ROST plane*, which appears as a mottled reflective network encircling small areas without a signal. These small signal gaps are caused from the cones above, which can be seen in two false-color composite images (*Fig 3*, 2[nd] row) of ROST (green) and IS/OS (magenta) and COST (red) signals. There is no relevant overlap between the cones and the rods (which would be visible as white or yellow signal overlap), as cones are entirely surrounded by rods in the depicted eccentricities. Contrary, a false-color composite image of the IS/OS+ (blue) and COST (red) plane in *Fig 3* demonstrates, that a majority of

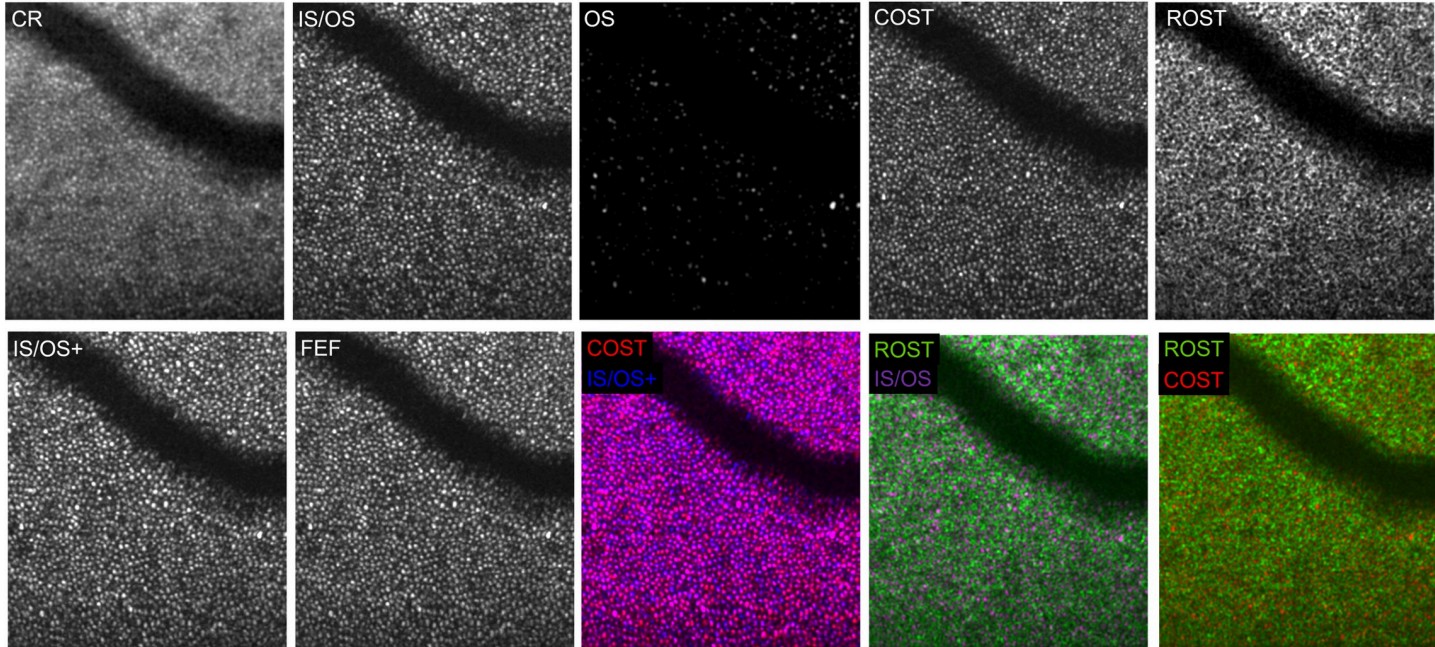

**Fig 3. The different photoreceptor planes visualized with AO-OCT at 6.5˚ eccentricity.** The complete retina (**CR plane**) image integrates signals from all retinal layers, beginning with the ILM through the choroid. This unsegmented depiction is comparable to two-dimensional AO-FC and AO-SLO imaging and shows the cone mosaic with reduced contrast and strong background signals, in particular from the rods and the RPE. The inner segment outer segment junction (**IS/OS plane**) shows a dense pattern of hyper-reflective dots. The cone outer segments (**OS plane**) show only a few hyperreflective and some several weak signals without a distinct pattern. The signals of the cone outer segment tips (**COST plane**) appear smaller and better separable then the IS/OS. The signals of the rod outer segment tips (**ROST plane**) form a mottled network. The **IS/OS+ plane** combines signals from the IS/OS to the OS. While its appearance resembles mostly to the IS/OS, subtle difference between these two en-face projections can be found in *S1 File*. The full en-face image (**FEF**) projects all signals from the IS/OS to the COST and gives the most complete appearance of the cone mosaic, especially when compared to the **CR plane**. Additionally, three false-color composite images are shown: A false color-composite image of the COST (red) and IS/OS+ plane (blue) shows signal overlap (purple) in most but not all areas and noticeable variations of signal strength between the layers (blue and red signals). In contrast, false-color composite images of the of the ROST (green) and the IS/OS (purple) or the COST (red) show no relevant overlap (white or yellow) between cone and rod signals.

signals between the two cone layers overlap (purple). However, only few cones show similar signal strengths in both layers (see bicolored purple-blue or purple-red dots). This indicates, that strength and size of signals within cones differ which is in agreement to previous reports and may change over time [23–25].

*Fig 3* also shows a full cone en-face image (*FEF plane*) which integrates signals from the IS/OS to the COST. In contrast to the CR image, where cone visibility is reduced due to signals from other layers, the *FEF plane* increases cone visibility and shows the most dense or complete depiction of the cone mosaic. This can also be seen in *S2 File,* where the FEF plane shows multiple signals, which are barely or not visible in the COST plane (images are taken from the same eye and region as *S1 File*). *S3 File* shows an en-face fly-through of the outer retina of a 23yo healthy woman, depicted with the AO-OCT system, beginning with the signals from the IS/OS plane and ending after the RPE. Regarding the different meridians, there was no remarkable difference between the 4 directions but there were distinctively visible differences in cone packing densities between ecc 6.5˚ and ecc 2.5˚ in all eyes (see *Fig 4*). A complete depiction of different cone planes in each meridian and eccentricity of a 26yo man (OS) can be seen in *S1 Fig*.

Inter-rater reliability of photoreceptor-counting using the Find Maxima function: The 60 images selected randomly for ICC assessment were composed from 20 images each of the IS/OS plane (of which 9 images were from ecc 2.5˚ and 11 images from ecc 6.5˚), COST plane (11

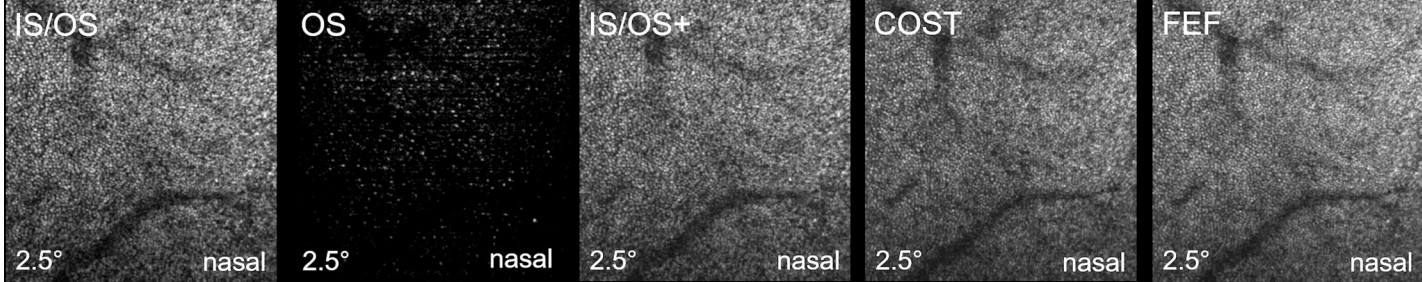

**Fig 4. The 5 different cone planes, depicted at 2.5˚nasal in a 26yo healthy male.** Inner segment outer segment junction zone (**IS/OS**) shows a dense mosaic of hyper-reflective dots. Cone outer segments (**OS**) show significantly less hyperreflective dots beside several blurry and weak signals. Combination of IS/OS and OS (**IS/OS+**) resembles mainly the IS/OS mosaic with some additional signals from the OS below (see also S1 File). Dots from the mosaic of the cone outer segment tips (**COST**) are better distinguishable compared to the IS/OS. Through integration of all cone signals to a full en-face image (**FEF**), the most complete appearance of the cone mosaic is achieved. S1 Fig gives a complete overview of all 8 regions imaged in this eye.

from ecc 2.5˚, 9 from ecc 6.5˚) and FEF plane (9 from ecc 2.5˚, 11 from ecc 6.5˚). Application of the Find Maxima function with rechecking showed excellent inter-rater reliability with ICCs above .99. Table 1 shows the mean density values and the ICCs for all three graders and three sets of images. The mean difference for cone density measurements between grader 1 (AR) and grader 2 (MS) were -117/mm$^2$ (range ±690) for IS/OS, +159/mm$^2$ (range -1.110 to +1.250) for COST and +201/mm$^2$ (range -420 to +830) for FEF. The mean difference for cone density measurements between grader 1 and grader 3 (VH) were +166/mm$^2$ (range -420 to +1.110) for IS/OS, -30/mm$^2$ (range -970 to +1.750) for COST and +152/mm$^2$ (range -550 to +970) for FEF. Values between grader 2 and grader 3 showed similar differences. In percentage, there was a range from ±0% to 10.1% for single measurements. The mean differences between graders were below 3% for all three planes.

Quantitative assessment of photoreceptor signals: 640 en-face images were assessed. The mean overall cone density at ecc 2.5˚ was 19.530/mm$^2$ (SD 2.17) at IS/OS, 19.920/mm$^2$ (SD 2.30) at IS/OS+, 20.450/mm$^2$ (SD 2.25) at COST and 21.160/mm$^2$ (SD 2.44) for FEF. The range was between 14.390/mm$^2$ (IS/OS) and 26.160/mm$^2$ (FEF). For ecc 6.5˚, corresponding values were 12.980/mm$^2$ (SD 1.46, IS/OS), 13.280/mm$^2$ (SD 1.49, IS/OS+), 13.410/mm$^2$ (SD 1.48, COST) and 13.810/mm$^2$ (SD 1.61, FEF) with a range from 9.830/mm$^2$ (ISOS) to 18.130/mm$^2$ (FEF).

Regarding the comparison of measured cone densities between each of the four layers, the linear models showed significant differences in all four meridians and both eccentricities (all p-values p≤.001) as why pairwise comparisons of all planes in each region were performed. Generally, the cone density was highest in FEF, followed by COST, IS/OS+ and finally IS/OS. These pairwise differences were statistically significant in most cases, as shown detailed in Table 2. The mean difference in cone density for IS/OS and IS/OS+ ranged between 48 and

**Table 1. Interclass correlation coefficients (two way mixed effect model, absolute agreement, three graders) for cone density measurements utilizing the Find Maxima function.** Three different sets of en-face images were used, each consisting of a random selection of 20 images.

| Grader | Cone Density /mm$^2$ | | | | | |
|---|---|---|---|---|---|---|
| | Random Set of IS/OS images (n = 20) | | Random Set of COST images (n = 20) | | Random Set of FEF images (n = 20) | |
| | Mean [SD] | Min Max | Mean[SD] | Min Max | Mean [SD] | Min Max |
| Grader 1 | 17.599 [4.263] | 9.830 23.670 | 17.315 [4.170] | 10.930 23.370 | 17.170 [3.646] | 12.600 23.940 |
| Grader 2 | 17.716 [4.290] | 9.970 23.390 | 17.156 [4.017] | 10.800 22.980 | 16.969 [3.662] | 12.600 24.010 |
| Grader 3 | 17.433 [4.256] | 9.550 23.110 | 17.285 [4.175] | 10.930 23.390 | 17.018 [3.587] | 12.610 24.080 |
| **ICC** | **.998** (95% CI: .996 - .999) | | **.995** (95%CI: .990 - .998) | | **.997** (95% CI: .994 - .999) | |

**Table 2. Mean differences of cone density between the IS/OS, IS/OS+, COST and FEF layers, shown for the 4 meridians and two eccentricities.** All numbers are given in cones/mm².

| Layer | Superior 2.5˚ | | | | Superior 6.5˚ | | | |
|---|---|---|---|---|---|---|---|---|
| | IS/OS | IS/OS+ | COST | FEF | IS/OS | IS/OS+ | COST | FEF |
| IS/OS | - | -48 | -505* | -1.218* | - | -235* | -484* | -720* |
| IS/OS+ | +48 | - | -457 | -1.170* | +235* | - | -249* | -484* |
| COST | +505* | +457 | - | -713* | +484* | +249* | - | -235* |
| FEF | +1.218* | +1170* | +713* | - | +720* | +484* | +235* | - |
| | Nasal 2.5˚ | | | | Nasal 6.5˚ | | | |
| IS/OS | - | -630* | -803* | -1.938* | - | -311* | -346* | -913* |
| IS/OS+ | +630* | - | -173* | -1.308* | +311* | - | -35 | -602* |
| COST | +803* | +173* | - | -1.135* | +346* | +35 | - | -567* |
| FEF | +1.938* | +1.308* | +1.135* | - | +913* | +602* | +567* | - |
| | Inferior 2.5˚ | | | | Inferior 6.5˚ | | | |
| IS/OS | - | -436* | -1.107* | -1.647* | - | -332* | -325* | -761* |
| IS/OS+ | +436* | - | -671* | -1.211* | +332* | - | +7 | -429* |
| COST | +1.107* | +671* | - | -540* | +325 | -7 | - | -436* |
| FEF | +1.647* | +1.211* | +540* | - | +761* | +429* | +436* | - |
| | Temporal 2.5˚ | | | | Temporal 6.5˚ | | | |
| IS/OS | - | -408* | -1.218* | -1702* | - | -360* | -588* | -948* |
| IS/OS+ | +408* | - | -810* | -1.294* | +360* | - | -228 | -588* |
| COST | +1.218* | +810* | - | -484* | -588* | +228 | - | -360* |
| FEF | +1702* | +1294* | +484* | - | +948* | 588* | +360* | - |

* Difference statistically significant, p < .05.

630/mm², for IS/OS+ and COST between 7 and 810/mm² and for COST and FEF between 235 and 1.135/mm². For better estimation, we calculated the relative mean difference as followed: The number of cones/mm² in the IS/OS plane was between 0.3% and 3% lower than IS/OS+, between 2.3% and 6.6% lower than COST and between 5.9% and 8.3% lower than FEF. This relationship can be seen in *Fig 5*, which shows the mean cone density for all 32 planes, grouped by layer, meridian and eccentricity.

Regarding the comparison of cone densities (FEF plane) for the four meridians and two eccentricities, the linear model showed a significant difference between the 8 regions (p<001), with the nasal meridian having the highest density, followed consecutively by the temporal, inferior and superior meridians. However, these differences did not reach statistical significance in all pairwise comparisons. At ecc 2.5˚, the difference in mean cone density between the nasal and the temporal (nasal: +55/mm²) and inferior (nasal: +436/mm²) meridian was not significant (p = .916 and p = .387). The superior meridian showed significantly lower densities than the three other meridians (difference between -3.737 and -4.173/mm²)(all p-values < .001). At ecc 6.5˚, the difference between the meridians was larger. Here, the cone density of the nasal meridian was significantly higher than for the temporal meridian (nasal: +969/mm²), inferior meridian (nasal: +2.298/mm²) and inferior quadrant (nasal: +3.052/mm²) (all p-values ≤.004). Additionally, the further differences in cone density (temporal > inferior > superior, differences ranging between 754/mm² and 2.083/mm²) were statistically significant (all p-values ≤.006).

As for the differences in cone densities between the two eccentricities, each layer in the four meridians showed significantly more cone signals at ecc 2.5˚ than at ecc 6.5˚, with mean differences ranging between 6.564/mm² (IS/OS) and 7.355/mm² (FEF)(all p-values < .001).

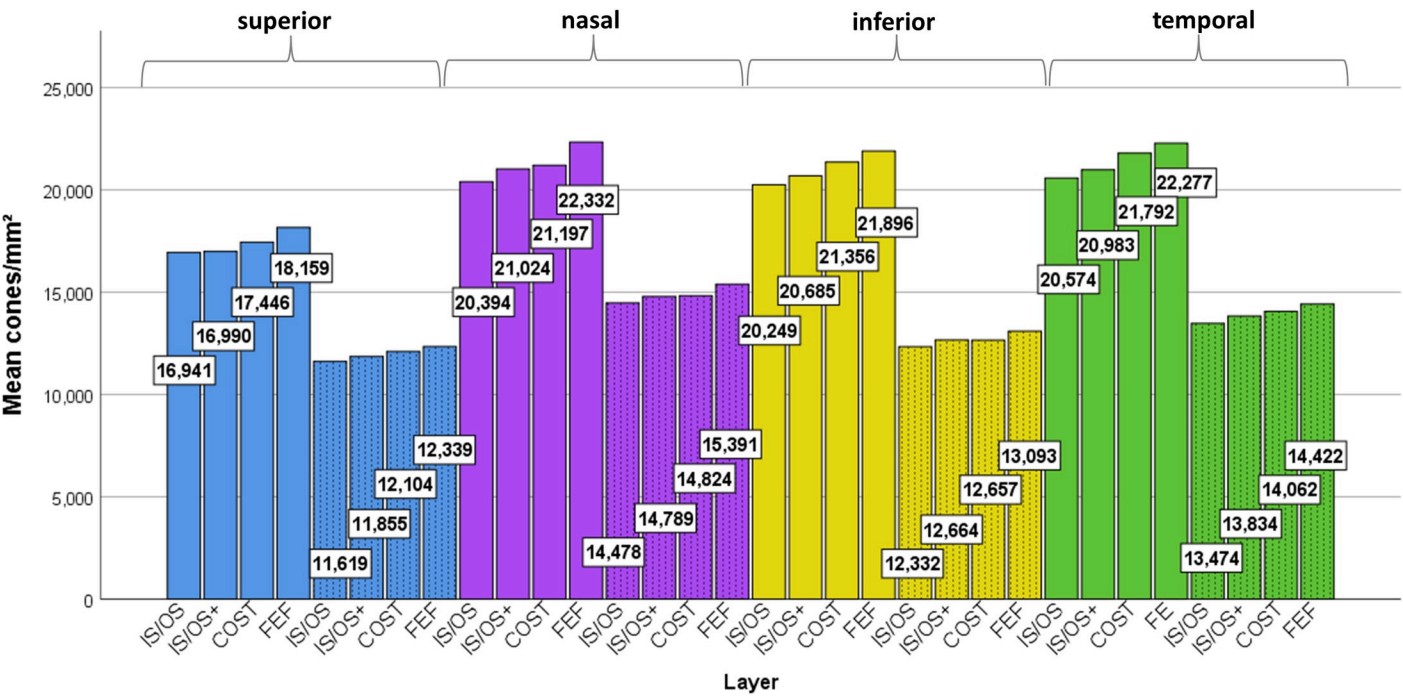

**Fig 5. Mean cone density of all 4 planes, 4 meridians (blue = superior, purple = nasal, yellow = inferior, green = temporal) and 2 eccentricities (solid fill = ecc 2.5˚, dashed fill = ecc 6.5˚).** N = 20 eyes. Density is generally higher at a foveal eccentricity of 2.5˚ compared to 6.5˚. In all 4 meridians and 2 eccentricities, density was highest in the FEF images, followed by COST, IS/OS+ and ISOS. Significant differences are marked in Table 2.

Regarding a general difference in cone density between the right and the left eye, no statistical difference was found at any layer, meridian and eccentricity.

## Discussion

This study assessed the signal composition and density of cone photoreceptors in healthy eyes with AO-OCT. So far, cone density values for healthy individuals have been reported in histologic examinations and studies using AO-FC and AO-SLO. However, histologic examinations are performed *ex-vivo* and the specimen is exposed to physical stress and tissue shrinkage, whereas two-dimensional AO imaging combines cone signals with distorting background signals from other retinal layers and cannot determine the definite origin of a signal.

One strength of AO-OCT lies in its ability to separate signals from the different layers of the photoreceptors. Subsequently, an important goal of this study was to provide more detailed information on the various cone layers in healthy individuals of varying age. In our participants, aged 23 to 67, AO-OCT imaging worked generally well, what was to be expected as we used rather tight inclusion and exclusion criteria. Regarding the depiction of the individual layers, the most distinct cone signals were found at the IS/OS and COST planes, which was true for every region in all our patients and is also in agreement with our previous clinical experience and studies introducing AO-OCT technology [22,24,27,28,32–36]. The cone signals at the COST plane showed a better differentiable appearance compared to the IS/OS plane, which still presented with a distinct mosaic pattern but the hyper-reflective dots did appear slightly fuzzy in direct comparison. This is interesting as the OCT beam passes through the IS/OS before it reaches the COST and the latter is in axial proximity to the signals from the ROST and the RPE. However, the less clear appearance of the IS/OS signal compared to the COST has been described previously and can be explained by variations of signal properties due to

waveguiding within the photoreceptors or by a possible influence of the rod inner outer segment junction, which is axially located at the same position as the cone IS/OS [19,34,37]. While we did not evaluate the difficulty of cone counting between the different layers statistically, all graders gave the feedback that analysis of the COST plane felt slightly easier than of the IS/OS. In opposite to the strong hyper-reflective signals of these two layers, the cone OS plane showed mainly weak signals and only a small number of hyper-reflective dots, which have also been found in technical feasibility studies of AO-OCT. These dots may be a result of defects within the packing density of cone outer segment discs [22,24,25,36].

The availability of a three-dimensional AO-OCT volume allows not only for separate assessment of the different cone layers but also to combine them for a better understanding of their qualitative and quantitative relationship. For this reason, we merged the signals from the IS/OS and OS to an IS/OS+ image and all signals of the cone layers to a full en-face image. While retinal diseases show different affections of the individual layers, for example COST being potentially more severely affected than IS/OS [28], one would expect identical mosaics at the level of IS/OS and COST and thus also in a full en-face image in healthy individuals. However, this is not the case, as differences in cone signals between the layers were not only observed qualitatively (compare *S1 File*, *S2 File*, *Fig 3*) but also quantitatively (compare *Table 2*, *Fig 5*). The full en-face image, which showed the most complete cone mosaic, had on average 3.1% more signals than the COST, 4.9% more signals than the IS/OS+ and 6.8% more signals than the IS/OS. This indicates, that the condition of a specific cone layer cannot be deduced from a full en-face -let alone a complete retina- image, and on the other hand, an overview of the total cone count cannot be gained from a segmentation of a single cone layer.

There are several causes that may contribute to the different cone densities between the layers. First, the strength and position of the signal can be affected by temporary changes in cone structures such as disc shedding, which has been shown to cause temporal signal loss in the COST by Kocaoglu et al. but can in some cases even produce multiple reflections [23,25,38]. Second, slight variations in cell morphology such as variation in width or axial positioning of cone segments but also changes in cone alignment due to subtle RPE irregularities can have an impact on their depiction [27,34,37]. Third, signal distortion from rod signals and other sources of blur could have an impact on the visualization of cones [19,34]. The slightly fuzzy, less separated appearance of the cone signals at the IS/OS plane, may lead to false low numbers, if some cone signals were not identified correctly by the graders. Misidentification of just a few signals can have a scaled impact, as the majority of studies regarding photoreceptor densities up to date, whether it being based on histology or AO-FC/SLO/OCT imaging, use very small areas, usually the size from 50x50 to 150x150μm or less, to calculate a density per mm$^2$. However, using larger sampling window sizes may also result in inaccurate assumptions, as the cone density shows a steady decrease with growing foveal eccentricity and there is a clearly visible change in cone packing density over the course of our AO-OCT images as well as AO-FC and SLO images [31,39].

Regarding the measurement of total cone density, current AO-FC and AO-SLO systems have two relevant benefits over AO-OCT systems: First, the images are less prone to motion artefacts due to faster en-face acquisition times; second, they often utilize image-averaging algorithms, improving the overall image quality and thus cone depiction [31,40–43]. The disadvantages are the unwanted collection of light from other retinal layers and the uncertain origin of a signal due to the limited axial information. Our results show that the mosaic pattern of the COST plane largely resembles the mosaic pattern of the full en-face image whereas projections from other retinal layers such as the ROST or RPE strongly reduce the depiction of cones signals as shown in the CR image, which is comparable to the images acquired with two-dimensional AO systems.

In this study, the FEF images showed the most complete and dense depiction of the cone mosaic. In these images, we found a cone density of 21.100/mm$^2$ at a foveal eccentricity of approximately 0.87mm (2.5˚). We also experienced a significant decline of cone signals with increasing eccentricity, as the cone density between ecc 2.5˚ and ecc 6.5˚ dropped between 6.500/mm$^2$ (IS/OS) and 7.300/mm$^2$ (FEF) or approximately 34%. These results are similar to histology and two-dimensional AO-FC and AO-SLO imaging.

In an *ex vivo* assessment of 8 donor eyes, Curcio et al. found a cone density of approximately 20.000/mm$^2$ at 1 mm foveal eccentricity [44]. Using different AO-SLO systems, Song et al. reported mean densities between 18.500 to 24.200/mm$^2$ at ecc 0.9 mm [45], Zhang et al. showed approximately 20 to 25.000 cones/mm$^2$ at ecc 1mm [31], and Park et al. found densities between 18.600 and 20.200/mm$^2$ at ecc 1mm [46]. Lombardo et al. used an AO-FC system and reported mean densities around 17.000/mm$^2$ at 1.1mm [39].

The comparison of the measured cone densities between the different AO approaches including the present study suggests, that there is generally a good level of agreement in cone density measurements between substantially different imaging methods. This is especially true considering that there are not only slight differences in determination and sizing of the sampling window or the correction for axial eye-length, but there is also a high level of inter-individual variation of measured cone density, which has been reported between 8.9% and 15% and was 11.6% in our study [44–47]. Regarding the effect of the main meridians on cone density, the cone densities of all layers in our subjects were highest at the nasal, then the temporal, then the inferior and finally the superior meridian. The difference was larger at ecc. 6.5˚ (approximately 2.1mm) than at ecc 2.5˚. While a lower eccentricity-dependent decline of cone density was described for the horizontal meridians in histology and AO-SLO (referred as "cone streak"), a statistical difference was so far only reported by Park et al. who found the same effect of meridians on cone density as we did (N,T > I, S) at ecc 1.0mm, but not at other eccentricities [44–46]. Although our results show a clear tendency, we believe that a larger sample size will be required to further validate the effect of meridians.

A further aim of our study was to utilize the "Find Maxima" function in the open-source image platform Fiji as an easy tool for semi-automated counting of cone density, which has been previously applied on AO-SLO images [31]. All graders, who were previously used to only manual counting, reported that the application of the "Find Maxima" feature with manual rechecking did substantially save time. Similar to AO-SLO, ICCs between graders were excellent and results of density measurements differed in most cases between 1.0% and 3% or roughly 150 to 500 cones/mm$^2$. Only in 3 of 180 comparisons between the graders, values above 8% were found with the highest differences being slightly above 10%, corresponding to approximately 1.500 cones/mm$^2$. Thus, despite being rare, the occasional appearance of outliers should be considered when interpreting cone density measurements. In comparison to inter-individual variability and the effect of eccentricity however, the clinical impact of these outliers appears to be clinically less relevant, especially as differences in cone density between healthy and diseased retinas have exceeded 10.000/mm$^2$ and even 15.000/mm$^2$ at same eccentricities [27,28].

This study has several limitations. Although the number of included eyes is similar to histologic studies and reports using AO-SLO/FC [31,39,44,45], Park et al. were able to assess the density in 192 eyes using AO-SLO [46]. Processing, segmenting and analysing of three-dimensional AO-OCT volumes is significantly more time and resource consuming than that of two-dimensional images and therefore the small number of subjects included is justifiable. While the aim of this study was to add information regarding the composition of cone layers in healthy individuals, the provided density values, variability and effects of eccentricity and meridians correspond well to the numbers provided by Park et al. [46]. Regarding the inclusion and exclusion of eyes, the criteria we used were similar to the ones of the referenced AO

studies. The AO-OCT system used in our study is not able to sufficiently depict single cones within the central degree of the fovea, where the highest number of cones is to be found. However, this limitation affects most AO-SLO and AO-FC systems [2,20,36,39,40,43,45,46]. Designing AO assisted imaging devices always requires compromises between achievable resolution and clinical usability, with our instrument being optimized for pupillary diameters of 5mm which can also be achieved in elderly individuals and most patients with ocular diseases.

In conclusion, the results of our study show that there are significant differences between the qualitative and quantitative depiction of IS/OS, OS and COST layers as well as the two composite cone layers IS/OS+ and FEF. For a comprehensive assessment of layer integrity, it is recommended to analyze each of the layers in conjunction with the others and refrain from basing conclusions solely on separate assessments of the IS/OS or COST layers. For comparison of overall cone densities, integrating AO-OCT signals from the cone layers to a FEF image can be used, with results similar to findings obtained by AO-SLO or AO-FC imaging.

## Supporting information

**S1 Fig. The 5 different cone planes, depicted at 2.5˚ and 6.5˚ foveal eccentricity in all 4 meridians of a 26yo healthy male.** There is no distinctive difference between the separate images of each meridian (compare each of the 4 images within every column). As expected from previous studies, there is a remarkable difference in density and size of the hyper-reflective dots between 2.5˚ (left half of the image) and 6.5˚ (right half of the image) foveal eccentricity.
(TIF)

**S1 File.**
(GIF)

**S2 File.**
(GIF)

**S3 File.**
(GIF)

## Author Contributions

**Conceptualization:** Adrian Reumueller, Michael Pircher.

**Data curation:** Adrian Reumueller, Lorenz Wassermann, Matthias Salas, Markus Schranz, Valentin Hacker.

**Formal analysis:** Adrian Reumueller.

**Investigation:** Adrian Reumueller, Lorenz Wassermann, Matthias Salas, Markus Schranz, Valentin Hacker.

**Methodology:** Adrian Reumueller.

**Project administration:** Andreas Pollreisz.

**Supervision:** Michael Pircher, Ursula Schmidt-Erfurth, Andreas Pollreisz.

**Validation:** Adrian Reumueller, Stefan Sacu, Wolfgang Drexler, Andreas Pollreisz.

**Visualization:** Adrian Reumueller.

**Writing – original draft:** Adrian Reumueller, Wolfgang Drexler, Ursula Schmidt-Erfurth.

**Writing – review & editing:** Adrian Reumueller, Lorenz Wassermann, Georgios Mylonas, Stefan Sacu, Michael Pircher, Andreas Pollreisz.

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
