## [Decision Letter · Decision Letter 0]

25 Nov 2020

PONE-D-20-28955

Three-dimensional composition of the photoreceptor cone layers in healthy eyes using adaptive-optics optical coherence tomography (AO-OCT)

PLOS ONE

Dear Dr. Reumueller,

Thank you for submitting your manuscript to PLOS ONE. After careful consideration, we feel that it has merit but does not fully meet PLOS ONE’s publication criteria as it currently stands. Therefore, we invite you to submit a revised version of the manuscript that addresses the points raised during the review process.

Both learned reviewers have provided substantive and constructive comments. A revised manuscript should aim to incorporate changes addressing the response to these comments. 

We look forward to receiving your revised manuscript.

Kind regards,

Sanjoy Bhattacharya

Academic Editor

PLOS ONE

Journal Requirements:

Reviewers' comments:

Reviewer's Responses to Questions

**Comments to the Author**

1. Is the manuscript technically sound, and do the data support the conclusions?

Reviewer #1: Yes

Reviewer #2: Yes

2. Has the statistical analysis been performed appropriately and rigorously? 

Reviewer #1: I Don't Know

Reviewer #2: Yes

3. Have the authors made all data underlying the findings in their manuscript fully available?

Reviewer #1: Yes

Reviewer #2: No

4. Is the manuscript presented in an intelligible fashion and written in standard English?

Reviewer #1: Yes

Reviewer #2: Yes

5. Review Comments to the Author

Reviewer #1: This Research Article named Three-dimensional composition of the photoreceptor cone layers in healthy eyes using

adaptive-optics optical coherence tomography. Statistical Analysis performed appropriately, All datas underlying the findings are available, The manuscript is written in standard English.

In Fig-1 comment, you mentioned a "23 yo healthy", I would like to follow the same format and mention the word "healthy" again in Fig-4 if it is the case. Also, I am wondering if you can upload a copy of the informed consent that you used in this study.

Reviewer #2: This is an interesting study on the use of AO-OCT to assess photoreceptor cones at different retina planes. I only have a few comments for the authors to consider:

“they are limited to a two dimensional view” change to “they are limited to a two dimensional en-face view”

“At least 3 volumes were acquired at each of the 8 regions. Volumes with the least amount of micro saccades were selected for further analysis.” How many volumes were analysed then, only the best one. This is not clear?

Are retinal layers segmented? Or the enface is generated at an approximate location. This was not clear and could be a potential source of error as the curve shape (or tilt) of the tissue can interfere in the measurement?. Please explain

Fig 2, would it be possible to show the entire B-scan as well as the zoomed with the layers of interest.

Table 1 has two nasal set of values, the inferior is missing.

The eccentricity information should be added in the label of fig 5, to facilitate reading of the graph.

I would recommend to move this section to the top of the results section, to first set the repeatability of the technique. “As for the inter-rater agreement, the 60 images selected randomly for ICC asses…” This values may be best in a table and the % should be also added in the table for the reader to assess the technique.

10.1% for single measurements is not that low, please comment on this large variability.

Correct this sentence “ layers, however. In opposite to the strong”

6. PLOS authors have the option to publish the peer review history of their article (what does this mean?). If published, this will include your full peer review and any attached files.

Reviewer #1: **Yes: **NAYEF K ALSHAMMARI

Reviewer #2: No

---

## [Author Response · Author response to Decision Letter 0]

29 Nov 2020

We want to thank the Reviewers and the Academic Editors for their constructive comments.

Please refer to the Author's Response/Cover Letter for detailed information.

---

## [Decision Letter · Decision Letter 1]

26 Dec 2020

Three-dimensional composition of the photoreceptor cone layers in healthy eyes using adaptive-optics optical coherence tomography (AO-OCT)

PONE-D-20-28955R1

Dear Dr. Reumueller,

We’re pleased to inform you that your manuscript has been judged scientifically suitable for publication and will be formally accepted for publication once it meets all outstanding technical requirements.

Kind regards,

Sanjoy Bhattacharya

Academic Editor

PLOS ONE

Additional Editor Comments (optional):

Reviewers' comments:

Reviewer's Responses to Questions

**Comments to the Author**

1. If the authors have adequately addressed your comments raised in a previous round of review and you feel that this manuscript is now acceptable for publication, you may indicate that here to bypass the “Comments to the Author” section, enter your conflict of interest statement in the “Confidential to Editor” section, and submit your "Accept" recommendation.

Reviewer #1: (No Response)

Reviewer #2: All comments have been addressed

2. Is the manuscript technically sound, and do the data support the conclusions?

Reviewer #1: Yes

Reviewer #2: Yes

3. Has the statistical analysis been performed appropriately and rigorously? 

Reviewer #1: I Don't Know

Reviewer #2: Yes

4. Have the authors made all data underlying the findings in their manuscript fully available?

Reviewer #1: Yes

Reviewer #2: Yes

5. Is the manuscript presented in an intelligible fashion and written in standard English?

Reviewer #1: Yes

Reviewer #2: Yes

6. Review Comments to the Author

Reviewer #1: This Research Article named Three-dimensional composition of the photoreceptor cone layers in healthy eyes using

adaptive-optics optical coherence tomography. Statistical Analysis performed appropriately, All datas underlying the findings are available, The manuscript is written in standard English.

Reviewer #2: The authors have addressed all my initial comments. I have no further comments to the authors.

7. PLOS authors have the option to publish the peer review history of their article (what does this mean?). If published, this will include your full peer review and any attached files.

Reviewer #1: **Yes: **NAYEF K ALSHAMMARI

Reviewer #2: No

---

## [Editor Report · Acceptance letter]

30 Dec 2020

PONE-D-20-28955R1 

Three-dimensional composition of the photoreceptor cone layers in healthy eyes using adaptive-optics optical coherence tomography (AO-OCT) 

Dear Dr. Reumueller:

I'm pleased to inform you that your manuscript has been deemed suitable for publication in PLOS ONE. Congratulations! Your manuscript is now with our production department. 

Kind regards, 

on behalf of

Dr. Sanjoy Bhattacharya 

Academic Editor

PLOS ONE